# Evolution of human leptospirosis in French Guiana, 2016–2022

Mathilde Zenou[1,2], Pascale Bourhy[3], Philippe Abboud[1], Mona Saout[4,5], Félix Djossou[1,5], Céline Michaud[6], Arsène Kpangon[7], Alexis Fremery[5,8], Nicolas Higel[9], Jean-François Carod[10], Christelle Prince[1], Sabine Trombert-Paolantoni[11], Alexia Barbry[12], Mathieu Nacher[5,13], Mathieu Picardeau[3], Loïc Epelboin[1,5,13], Paul Le Turnier[1,5,13]*

1 Department of Infectious and Tropical Diseases, Cayenne University Hospital, Cayenne, French Guiana, 2 Department of infectious and Tropical Diseases, Toulouse University Hospital, Toulouse, France, 3 Biology of Spirochetes Unit - French National Reference Center for Leptospirosis - WHO Collaborating Center for Reference and Research on Leptospirosis, Institut Pasteur, Paris, France, 4 Tropical Biome and Immuno Pathophysiology Laboratory (TBIP), University of French Guiana, Cayenne, French Guiana, 5 UA 17 Santé des Populations en Amazonie, Cayenne University Hospital, Cayenne, French Guiana, 6 Remote Prevention and Care Centers, Cayenne University Hospital, Cayenne, French Guiana, 7 Infectious Diseases Unit, Kourou University Hospital, Kourou, French Guiana, 8 Emergency Department, Cayenne University Hospital, Cayenne, French Guiana, 9 Intensive Care Unit, Cayenne University Hospital, Cayenne, French Guiana, 10 Clinical laboratory, Centre Hospitalier de l'Ouest Guyanais, Saint-Laurent-du-Maroni, French Guiana, 11 Microbiology Department, Cerba Laboratory, Frépillon, France, 12 Immunology Department, Biomnis Laboratory, Eurofins Biomnis, Lyon, France, 13 CIC INSERM 1424, Cayenne University Hospital, Cayenne, French Guiana

* paul.leturnier@gmail.com

## Abstract

### Background

Leptospirosis is a re-emerging zoonotic disease. In French Guiana (FG), updating its epidemiology is essential to guide prevention strategies. This study aimed to describe human leptospirosis cases from 2016 to 2022 and compare them to the cases from 2007 to 2014 (using similar diagnostic criteria).

### Methodology/Principal findings

A multicentric cross-sectional study was conducted in the three hospitals of FG and the remote health centers. Cases were identified via biological diagnostics, defined by a compatible clinical picture and a positive biological test and classified according to the test as confirmed (positive PCR or Microscopic agglutination test [MAT] ≥400) or probable (MAT=200 or positive IgM only without alternate diagnosis). Severe cases involved renal, circulatory, or respiratory failure, or death. A total of 188 cases were included, of which 138 (73.4%) were confirmed. Median (IQR) age was 38 (28–52) years, with a male-to-female ratio of 3.1. Twenty-six (13.8%) cases were severe, including 4 deaths (2.1%). Most patients had multiple exposure factors with rodent exposure being the most common; 67.5% (available data) had both domestic and non-domestic exposures. Notably, over a third of patients were in a situation of

**Data availability statement:** The datasets generated and/or analyzed during the current study are not publicly available due to ethical and legal restrictions, as they contain potentially sensitive personal health information. Data sharing is subject to approval by the French data protection authority (Commission Nationale de l'Informatique et des Libertés, CNIL). Upon reasonable request and pending authorization from CNIL, access to de-identified data may be granted. Requests for data access should be directed to the institutional data access committee via the Direction de la Recherche Clinique et de l'Innovation (DRCI) of Cayenne University Hospital, at: drci.promotion@ch-cayenne.fr. For regulatory matters, CNIL can be contacted at: Commission Nationale de l'Informatique et des Libertés (CNIL) 3 Place de Fontenoy – TSA 80715 – 75334 PARIS CEDEX 07, France Website: https://www.cnil.fr.

**Funding:** The author(s) received no specific funding for this work.

**Competing interests:** The authors have declared that no competing interests exist.

precarity, uninsured or lived in informal settlements. The mean (SD) annual number of cases increased from 5.8 (2.7) per 100,000 of the adult population in the 2007–2014 period to 14.0 (9.2) in the recent study period (p = 0.03). Early clinical suspicion improved, while the proportion of severe cases remained stable.

## Discussion/Conclusion

Leptospirosis is an increasing public health issue in FG which particularly affects socioeconomically vulnerable populations. Routes of transmission appear multiple.

## Author summary

Leptospirosis is a bacterial disease transmitted from animals, especially rodents, that can cause serious illness or death. In French Guiana, many people live in vulnerable conditions, such as informal settlements, with exposure to flooding and diverse animal reservoirs, making it important to update our knowledge of who is affected and how the disease spreads. This study updates the characteristics of patients diagnosed with leptospirosis from 2016 to 2022 and compares them with those from 2007–2014 using a similar and robust case definition. The overall burden of the disease, including severe and non-severe cases, doubled over the study periods. Significant changes included higher comorbidities, more patients born in Haiti, differences in occupation, and faster clinical suspicion. A trend toward older age and a higher proportion of female patients were also observed. Patients often live in precarious conditions and have multiple sources of exposure both at home and elsewhere. The proportion of severe cases remained similar. These findings highlight subtle shifts in the epidemiology of leptospirosis in French Guiana and provide insights to support more targeted public health interventions and the identification of at-risk populations.

## Introduction

Leptospirosis is a worldwide bacterial zoonotic disease, caused by pathogenic *Leptospira* spp., with around 1,000,000 cases per year and 58,000 deaths estimated in a study published in 2015 [1]. The intertropical zone accounts for 73% of worldwide cases. Leptospirosis is also a major cause of morbidity, estimated at 2.9 million of disability-adjusted life years in 2015, comparable to diseases such as leishmaniasis or schistosomiasis [2].

French Guiana (FG) is a French overseas territory located on the northeastern coast of South America mainly covered (>90%) by Amazonian rainforest. However, most of its ~300,000 inhabitants live in urban or suburban areas along the Atlantic coast. Half of the population lives under the French poverty threshold, often in informal settlements with unsanitary environmental conditions.

Until the study that assessed the human leptospirosis epidemiology in FG between 2007 and 2014, the disease was considered anecdotal [3–4]. Since then, many changes occurred that could potentially favor leptospirosis transmission. Indeed, recent climatic conditions suitable for leptospirosis emergence have been reported such as unprecedented rainfall associated with flooding in 2021 and 2022 [5]. In addition, a demographic increase in informal settlement especially around Cayenne, the main city of FG, has been reported during the last decade [6]. By late 2022, it was estimated that around 18,500 people were living in informal settlements along French Guiana's coastline. Moreover, an increase of 1.8% of informal settlement building has been reported between 2015 and 2019 [7]. This phenomenon occurred in a context of increasing precarity among FG inhabitants as well as continuous migratory flow from the neighboring countries (Brazil, Suriname and Guyana), but also Haiti, and Syria, Morocco and Afghanistan. In particular, the social, housing and residential characteristics of the French Guianese population have been associated with high levels of precarity and specific health issues [8] that could increase the risk of leptospirosis for local population and impact its management.

The transmission routes of leptospirosis and its epidemiology depend on numerous human, animal, and environmental driving factors which may vary over time. Yet, data on exposure factors for human leptospirosis are still very limited in FG. Moreover, the high biodiversity in French Guiana is likely to support a wide array of animal reservoirs, not limited to urban rodents. We hypothesize that the epidemiology of leptospirosis in French Guiana is complex and dynamic. Therefore, an update of the characteristics of the human leptospirosis cases - especially exposure factors- is mandatory to properly assess its burden and better understand the routes of transmission.

The objective of the study was to assess the human leptospirosis burden in French Guiana for the 2016–2022 period and measure the epidemiological changes since the 2007–2014 period.

## Methods

### Ethics statement

The institutional review board at each site approved the protocol. Each participant was informed by letter of the possibility to oppose the utilization of his or her data in the study. The study was approved by the CERMIT (Comité d'Ethique de Recherche en Maladies Infectieuses et Tropicales) Ethics Committee (number 2023-0702-2). Formal consent was not required for this type of observational retrospective study. All statistical analyses were performed on an anonymized dataset and procedures were in accordance with the ethical standards of the National Research Committee and the Declaration of Helsinki.

### Study design

This multicentric cross-sectional study was conducted in the three hospital centers of FG: Cayenne (CHC – Centre Hospitalier de Cayenne), Kourou (CHK – Centre Hospitalier de Kourou) and St Laurent du Maroni (CHOG – Centre Hospitalier de l'Ouest Guyanais) and in the Remote Prevention and Care Centers (RPCC). All three hospital centers had an emergency department. The number of acute care beds was 240 (including pediatrics and emergency short-stay hospital beds) at the CHC in 2021, and 28 and 30 (not including pediatrics and short-stay beds) at the CHK and the CHOG respectively in 2022; before 2018 the CHOG had 20 adult acute care beds. STROBE checklist is given in supplementary (S1 Checklist STROBE Checklist).

The primary objective was to assess the number of confirmed and probable leptospirosis cases in French Guiana during the study period. The secondary objectives were to describe the population demographics, leptospirosis reported risk factors, delays in care, biological technique used for diagnosis, infecting serogroups, clinical presentation, occurrence of severe forms, and to compare the main epidemiological characteristics with a previous study from the 2007–2014 period.

## Biological diagnosis, case definitions and inclusion criteria

During the study period, the biological diagnosis of leptospirosis was based on a compatible clinical picture, confirmed by biological tests. The choice of diagnosis test for leptospirosis was based determined by the attending clinician. In particular, PCR testing could be requested by clinicians on the sample type of their choice, according to their clinical judgment, as no standardized protocol was in place within the hospitals during the study period.

PCR diagnosis was based on an in-house real-time PCR targeting the 16S RNA coding gene and *lipL*32 gene. PCR [9–10] was performed on 200 µL of plasma, serum, or whole blood using the MagNA Pure 96 system. DNA extracts were amplified with primer pairs targeting 16S rRNA, *lipL*32, and an internal control (Phage M13), with probes labeled FAM/TAMRA or VIC/TAMRA. Reactions were run for 45 cycles on ABI7500 or QS5 thermal cyclers. For urine, 2 mL were centrifuged (12,000 rpm, 30 min), 0.5 mL with pellet was retained, and the same protocol applied. All routine PCR were performed by Biomnis laboratory.

Serological diagnosis of anti-*Leptospira* immunoglobulin M (IgM) was performed in laboratory Cerba or Biomnis using an enzyme-linked immunosorbent assay (ELISA) (Serion, Germany) [11–12]. Only positive tests were considered for this study. An equivocal test was considered as negative [13]. When the IgM titers were available in the laboratory database, only titers above a significant threshold of 50 IU/ml were defined as positive.

The Microscopic Agglutination Test (MAT) was performed in routine by BIOMNIS lab using a standard panel not specifically designed for French Guiana. This panel contained 11 serovars: Icterohaemorrhagiae, Copenhageni, Canicola, Australis, Pyrogenes, Pomona, Autumnalis, Grippotyphosa, Castellonis, Sejroe and Patoc. Sera analyzed by MAT were tested individually against each strain of the antigenic panel. The presumptive infecting serogroup was defined by the highest MAT titer detected in the latest serum sample. If the MAT titers for multiple serogroups were equal, the presumptive infecting serogroup was considered as indeterminate (coagglutinins).

For the study, the diagnosis of leptospirosis was confirmed if there was either a positive PCR in a biologic sample (blood, urine or cerebrospinal fluid) or a positive MAT titer of 1:400 or higher [3,14–17] (except for Patoc) as previously reported. The diagnosis of leptospirosis was considered probable in the case of MAT positive titer of 1:200 (except for Patoc). Patients with positive anti-*Leptospira* IgM above 50 IU/ml alone (without positive MAT or PCR) were considered as probable cases [18] only if no differential diagnosis was found in the medical records, whether IgM seroconversion occurred or not. Patients with positive IgM levels alone (without MAT or PCR criteria) and with a proven alternate diagnosis were not included to limit the inclusion of subjects with false positive IgM.

Finally, the inclusion criteria were having a certain or probable leptospirosis diagnosis between January 1, 2016, and December 31, 2022 and an age > 15 years at diagnosis.

Exclusion criteria were a missing medical file, opposition expressed by the patient to participate in the study and age < 18 years at the time of the data collection.

**Ancillary analysis *on the* 2021–2022 period.** An ancillary analysis was conducted using the available biological samples from cases managed at the Cayenne hospital between December 1, 2021, and June 1, 2022. They were systematically sent to the NRCL to perform molecular and serological identification. Amplification of the *lfb1* gene by real-time PCR and Sanger sequencing were performed to determine the species and, if possible the species-groups as previously described elsewhere [19]. The serological identification was performed by MAT at the NRCL on all available serum samples with positive IgM diagnosed during this specific period [20]. The serovars tested in the panel were Australis, Autumnalis, Ballum, Bataviae, Canicola, Celledoni, Cynopteri, Djasiman, Grippotyphosa, Hardjo, Hebdomadis, Icterohaemorrhagiae, Copenhageni, Javanica, Louisiana, Mini, Panama, Patoc, Pomona, Pyrogenes, Sarmin, Sejroe, Shermani and Tarassovi.

## Data collection

The results of biological diagnosis during the study period for the participating centers were screened to assess whether patients were eligible for this study.

Patients' electronic records were then screened for inclusion criteria. Once included, patient's data was retrospectively collected from electronic records in a standardized case report form. The data collected included demographic, exposure, clinical, biological and outcome information (see S1 Table). The suspicion of leptospirosis was considered at first medical contact if it was explicitly noted in the record. The delay of diagnosis suspicion was defined as the time between the first medical contact and the prescription of the biological test for leptospirosis or the mention on the medical chart of leptospirosis (whichever occurred first). Exposure investigation at first medical contact was defined as investigation of at least exposure to both rodents and fresh water. Exposures not mentioned in the patient file were considered as missing data. Domestic exposure was defined as exposure occurring inside or around the home, including areas such as the private garden.

Data from the census of the National Institute of Statistics and Economic Studies (INSEE)-which estimates the annual population in each French department- were used for the calculation of annual study cases ratio above the adult population of French Guiana (aged above 15 years) of the corresponding year [21]. The census of 2022 was not released so the annual population of this year was estimated with linear regression based on the previous years.

A paradoxical reaction (Jarisch-Herxheimer reaction [JHR]) was defined to standardize data collection as hypotension (systolic pressure < 90 mmHg or mean arterial pressure < 65 mmHg or a drop of 20 mmHg in systolic pressure if known hypertension) or respiratory distress in the 12 hours after antibiotic administration. The JHR was only collected when happening outside the intensive care unit (ICU). A severe form of leptospirosis was defined as either the use of vasopressors agent for hemodynamic support, mechanical ventilation, renal replacement therapy or death. Health insurance status was assessed at the time of the first consultation. Health insurance was defined as benefiting from either social security (accessible to foreign nationals with residence permits or French nationals), or state medical assistance (accessible to foreign nationals without a residence permit). β-lactams (except aztreonam), cyclins, macrolides, fluoroquinolones and aminoglycosides were considered as active antibiotic therapy against leptospirosis [22–23].

## Statistical analysis

Quantitative variables were described as mean and standard deviation or median and interquartile range, depending on the variable distribution, and qualitative variables were described with frequency and percentage with n/N to indicate the number of available data for each categorical variable with missing data. No imputation was used to address the missing data.

Given the exceptional rainfall observed in 2021–2022 in French Guiana, severe floodings occurred in areas of informal settlements. Thus, we evaluated whether two proxy indicators of precariousness—housing type and health insurance coverage—differed during 2021–2022 compared to 2016–2020.

The main characteristics of cases from the 2016–2022 were compared with those from the cases of the 2007–2014 period; a Student's t-test was used to compare numeric variables if the conditions of applicability were met using Shapiro-Wilk and Levene tests; a chi-squared test or Fisher's exact test was used for proportion comparison as appropriate. A p-value of 0.05 was considered significant. All tests were two-tailed.

All analyses were performed using the STATA 14 software (College Station, TX: StataCorp LLC).

## Results

During the 7-year study period 188 patients were included, 138 (73.4%) were confirmed cases (Fig 1).

### Distribution of cases

During the study period the mean annual number of included cases was 26.0 +/- 5.3 per year corresponding to 14 (+/-9.2) per 100 000 inhabitants aged >15 years old cases per year. The number of study cases per 100,000 of the population varied from 1.5/100,000 inhabitants in 2016 to 21.3/100,000 inhabitants in 2021 (Fig 2). Almost as many cases occurred

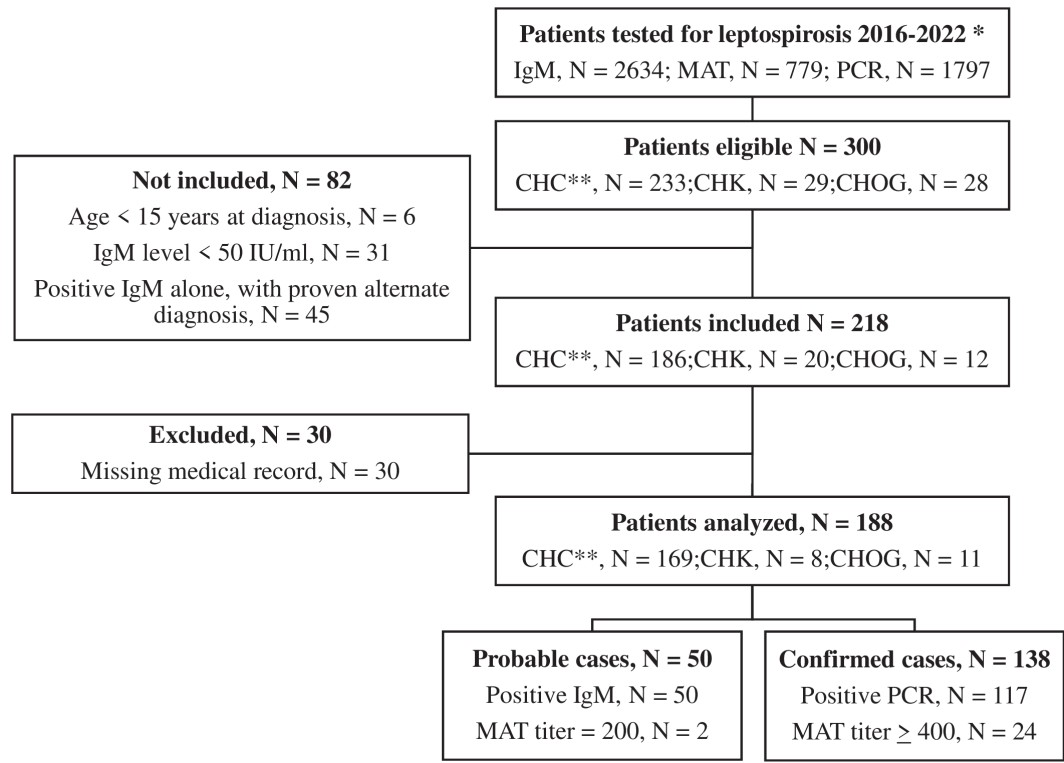

**Fig 1. Flow chart.** * Number of tests performed for the laboratories during the whole study period for of CHC (Centre Hospitalier de Cayenne) and during the 2018-2022 period for CHK (Centre Hospitalier de Kourou) and CHOG (Centre Hospitalier de l'Ouest guyanais) (patients could have repeated tests). ** Patients who first consulted in Remote Prevention Care Center were included in CHC tests results. IgM: Immunoglobulin M; PCR: Polymerase Chain Reaction; MAT: Microscopic Agglutination Test.

in 2021–2022 (n = 86) as in 2016–2020 (n = 91). Most patients initially consulted in CHC (n = 155, 82.5%), 8 patients (4.3%) in CHK, 9 (4.8%) in CHOG and 16 (8.5%) in remote prevention and care centers.

## Sociodemographic characteristics in the study population

The median age was 38 (28–52) years. Male to female sex ratio was 3.1. Sixty-seven patients (35.6%) had at least one comorbidity, mostly high blood pressure (n = 30, 16.0%) and diabetes (n = 17, 9%). Most patients were born in Haiti (n = 49/168, 29.2%), followed by Brazil (n = 44/168, 26.2%) (Table 1).

At the first hospital visit, 38.8% of patients had no health insurance (n = 66/170, 38.8%). Cases in 2021 and 2022 lived slightly more frequently in informal settlements (21/52,40.4%) than in 2016–2020 (13/43, 30.2%) (no significant difference) and had health insurance less frequently (35/81,43.2%) than in 2016–2020 (31/89, 34.8%) (no significant difference).

## Exposure factors in the study population

A wide variety of occupational exposures was reported in 72/154 (46.8%) patients: mostly building work (21/72, 29.1%) and gold mining (12/72, 16.7%). Garbage collecting was only reported once.

The main reported exposure factors were the proximity with rodents and freshwater contact followed by forest hiking, living in informal settlements, and proximity with non-rodent mammals (Table 2). Overall, most patients in whom several exposure factors were investigated (84/138, 60.9%) had at least two reported exposure factors.

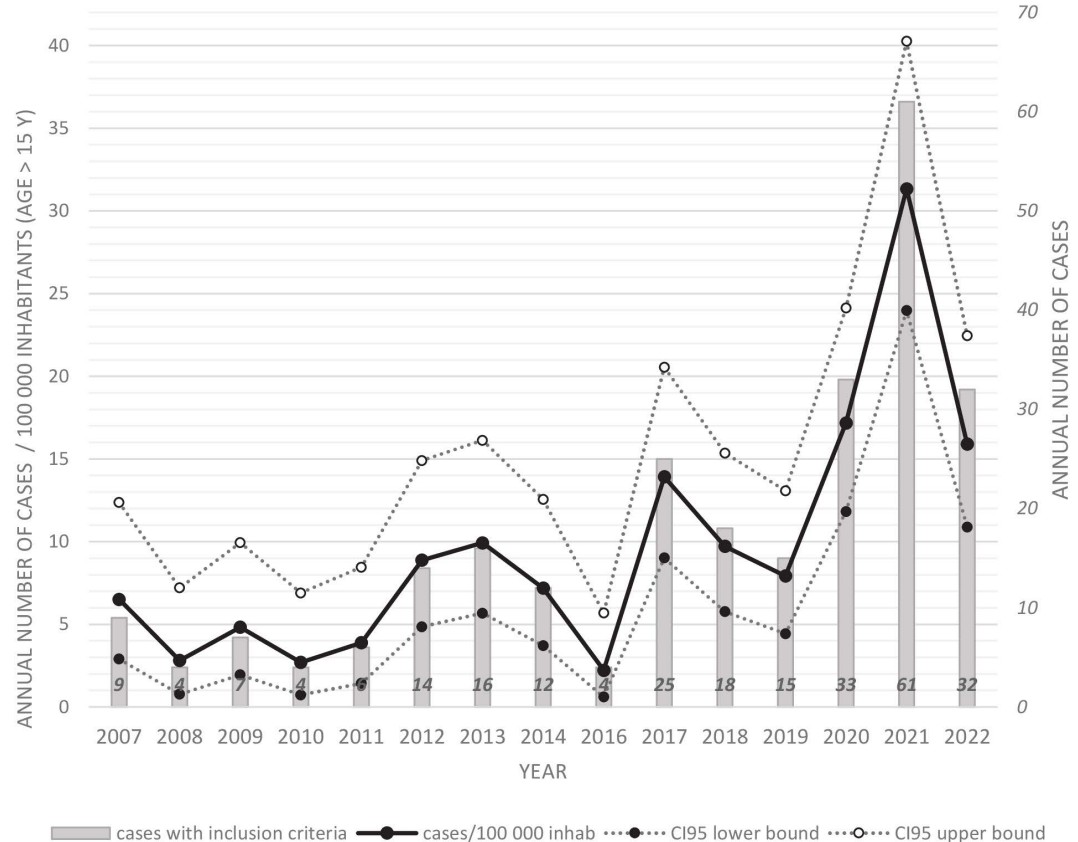

**Fig 2. Annual distribution of confirmed and probable study cases of human leptospirosis per 100,000 of the French Guiana adult population, 2007 - 2022\*.** \* a ratio of the number of cases with inclusion criteria above the French Guiana adult population (age > 15) was calculated with data extracted from INSEE census for each year of the study period [13]. The population of year 2022 was estimated with linear regression. Cases of the 2007-2014 period were reported from a previous study.

Concerning the exposure site, forty-four patients (65.7%) reported both domestic and non-domestic exposures (without accounting occupational exposure).

## Clinical characteristics and management of study population

The clinical characteristics of the study patients are shown in Table 3.

Arthromyalgia, fever and headache were the most frequently reported symptoms on admission. Jaundice, dyspnea and crackles on chest auscultation were frequently observed during follow-up.

The median duration between onset of the symptoms and first medical contact was 3.7 days (2.5-5.6, range 0–15). Regarding clinical management, leptospirosis was suspected upon hospital admission for 101 patients (53.7%). Median time between first medical contact and diagnosis suspicion was 12.2 (5.1-44.7) hours. Among patients with bleeding signs (N = 29), 10 (34.9%) were categorized as having severe leptospirosis and 5/29 (17.2%) underwent mechanical ventilation.

One hundred and forty-seven patients (78.2%) were hospitalized for at least one night; 9 out of 16 patients initially seen in remote centers were hospitalized - mostly in CHC. Twenty-eight patients (14.9%) were admitted to ICU. Overall, 26 patients (13.8%) were classified as severe: 21 (11.2%) required vasopressor drugs, 8 (4.3%) required mechanical ventilation, 14 (7.5%) needed renal replacement therapy and 4 (2.1%) died in hospital.

**Table 1. Socio-demographic, living and health status characteristics of patients with leptospirosis in French Guiana, 2016-2022.**

|  | Results |
|---|---|
| Age, years (N = 188) | 38 (28-52) |
| Age categories (N = 188) |  |
| 15-19yo | 8 (4.3) |
| 20-29yo | 44 (23.4) |
| 30-39yo | 50 (26.0) |
| 40-49yo | 29 (15,4) |
| 50-59yo | 28 (14.9) |
| 60-69yo | 23 (12.8) |
| ≥70yo | 6 (3.2) |
| Male gender (N = 188) | 142 (75.5) |
| Underlying condition (N = 188) | 67 (35.6) |
| Declared alcohol consumption (N = 121) | 31 (25.6) |
| Declared tobacco consumption (N = 133) | 47 (35.3) |
| Declared crack cocaine consumption (N = 93) | 9 (9.7) |
| Place of birth (N = 168) |  |
| Haiti | 49 (29.2) |
| Brazil | 44 (26.2) |
| French Guiana | 32 (19.1) |
| France (other than FG) | 19 (11.1) |
| Guyana | 7 (4.2) |
| Dominican Republic | 6 (3.6) |
| Suriname | 5 (3.0) |
| Other | 6 (3.6) |
| Length of stay in FG, years* (N = 104) | 11.3 (13.1) |
| < 1 year | 13 (12.0) |
| 1-5 years | 37 (34.3) |
| > 5 years | 58 (53.7) |
| Absence of health insurance (N = 170) | 66 (38.8) |
| Living in informal settlement (N = 95) | 34 (35.7) |

Data are mean (standard deviation) or median (IQR) for quantitative variables and n (%) for categorical variables. In case of missing data, n/N is specified for categorical variables and N is provided for quantitative variables

FG: French Guiana; yo: years old

* if born outside of French Guiana

Most patients received antibiotic therapy (N = 166, 89.5%), almost all regimens (99.2%) consisted of antibiotic theoretically active against *Leptospira*. The antibiotics administered were mainly third-generation cephalosporins (130/166, 78.3%), penicillins (66/166, 39.8%), doxycyclin (79/166, 47.6%) and other antibiotics (56/166, 33.7%) mainly aminoglycosides and macrolides. The median duration between onset of symptoms and antibiotic therapy initiation was 4 days (3–6). The median duration of antibiotic therapy was 7 days (6–10).

A JHR fulfilling our predefined criteria occurred in 15/165 (9.1%) patients who received antibiotics and had detailed post-antibiotic early follow-up data. Twenty-seven percent (7/26) of (26.9%) patients eventually classified as severe leptospirosis developed a JHR prior to ICU admission compared to six percent (8/139) of non-severe patients (odds ratio of 5.9 [95% confidence interval 1.6-21.3, p = 0.003]).

**Table 2. Exposure factors of patients with leptospirosis in French Guiana, 2016-2022.**

| | Results, N (%) |
|---|---|
| **Occupational exposure** (N = 154) | |
| No occupation reported | 38 (24.7) |
| Building work | 21 (13.6) |
| Gold mining | 12 (7.8) |
| "job" (informal work) | 11 (7.1) |
| Other at-risk occupation* | 28 (18.2) |
| Other non-at-risk occupation | 35 (22.8) |
| Retired | 9 (5.8) |
| **Non-occupational exposure** | |
| Proximity with rodents | 72/107 (67.3) |
| Freshwater contact | 69/79 (87.3) |
| Forest hiking | 41/96 (42.7) |
| Living in informal settlement | 34/95 (35.8) |
| Proximity with non-rodent mammals | 33/67 (49.3) |
| Gardening | 11/13 (84.6) |
| Proximity to waste | 3/7 (42.9) |
| Proximity to wastewater | 2/6 (33.3) |
| Flooding | 0/11 (0) |
| **Site of exposure** | |
| ≥ 1 domestic exposure | 89/102 (87.3) |
| ≥ 1 non-domestic exposure | 76/87 (87.4) |
| Both domestic and non-domestic | 44/67 (65.7) |
| Residence in urban area | 149/171 (87.1) |
| **Number of reported exposure factors** (N = 138) | |
| 1 | 54 (39.1) |
| 2 | 37 (26.8) |
| ≥3 | 47 (34.1) |

*farmer (n = 1), fisherman (n = 5), plumber (n = 1), woodcutter (n = 1), butcher (n = 1), gardener (n = 5), cook (n = 7), military (n = 5), canoeist (n = 1), garbage collector (n = 1)

### Biological characteristics and diagnosis of the study population

Biological characteristics of the patients are shown in S2 Table.

A PCR test was performed for 143 (76.1%) of 188 study patients. Among them, the most frequent sample was blood for 137 patients (95.8%), positive for 96 cases, followed by urine for 61 patients (42.7%), positive for 51 cases, and cerebro-spinal fluid for 6 patients (4.2%), positive for 2 cases. Both blood and urine PCR assays were performed in 55 patients (38.5%). The IgM ELISA test was performed in 136/188 patients (72.3%) with 87 (64.0%) positive results.

As routine diagnosis, 35/188 (18.6%) patients were tested with MAT. The median delay between symptoms onset and first MAT was 9.5 [28–52] days; 24/35 (68.6%) MAT were positive. A second test was performed for 4/35 patients, after 7, 19 and 20 days since first MAT sampling (1 missing data). Three patients were already MAT positive and one sero-converted from a negative MAT to a titer of 1:3200. Overall, among the 25 positive MAT, the median titer was 1:1600 [1:800–1:3200].

The ancillary study provided complementary MAT results for 19 patients. Two of them already had a MAT in routine care. One seroconverted from 0 to a titer of 1:12800 after 20 days. The second had a positive MAT on both tests but a distinct

**Table 3. Clinical characteristics of patients with leptospirosis in French Guiana, 2016-2022*.**

|  | On first exam, N (%) | During follow-up * **, N (%) |
|---|---|---|
| Fever *** | 95/174 (54.6) | – |
| Headache | 114/187 (61.0) | 5/63 (7.9) |
| Arthromyalgia | 127/183 (69.4) | – |
| Skin rash | 5/187 (2.7) | 6/142 (4.2) |
| Jaundice | 37/186 (19.9) | **20/109 (18.4)** |
| Conjunctivitis | 19/147 (12.9) | 6/134 (4.5) |
| **Respiratory signs** |  |  |
| Cough | 41/183 (22.4) | 5/111 (4.5) |
| Dyspnea | 27/187 (14.4) | **21/122 (17.2)** |
| Crackles at examination | 31/183 (16.9) | **17/114 (14.9)** |
| **Digestive signs** |  |  |
| Diarrhea | 51/122 (41.8) | 7/51 (13.7) |
| Nausea- vomiting | 69/186 (37.1) | 3/87 (3.5) |
| Abdominal pain | 56/186 (30.1) | 7/95 (7.4) |
| Defensive abdomen | 3/185 (1.6) | 1/145 (0.7) |
| **Neurological signs** |  |  |
| Neck stiffness | 13/125 (10.4) | 1/58 (1.7) |
| Other neurological alteration**** | 9/181 (14) | 11/134 (8.2) |
| Hemoptysis | 12/187 (6.4) | 7/134 (5.2) |
| Other bleeding manifestations | 3/187 (1.6) | 7/142 (4.9) |

*includes every sign appearing after admission, without pre-defined follow-up assessment

** signs appearing for more than 15% of population are in bold

*** fever documented within the 12 hours following admission

****confusion, nerve palsy, peripheral neuropathy, encephalopathy, seizure

n/N is given; N is patients with available data.

presumptive serogroup: Pomona on the 1st MAT 1:400 and Icterohaemorrhagiae on the second MAT with titer 1:800. The details of the leptospirosis biological diagnosis of all the patients included is provided in supplementary S3 Table.

The results obtained from the NRCL ancillary analysis (genomic and serological) with the routine MAT data were aggregated to describe the serogroups. Overall, a serogroup identification was obtained for 43 subjects. The identified serogroups were Icterohaemorrhagiae for 33 patients (77%) (seven with serovar Copenhageni, 15 with serovar Icterohaemorrhagiae and 11 with *lfb1* sequencing in favor of serogroup Icterohaemorrhagiae), Canicola for 7 patients (16%) and Pyrogenes, Ballum and Panama 1 patient per each (2%). Nine had coagglutinins, including two with *lfb1* sequencing in favor of *L. interrogans* serogroup Icterohaemorrhagiae. *lfb1* sequencing result was considered over MAT performed at NRCL in a discordant case. In this case, *lfb1* gene sequences analysis was in favor of *L. interrogans* serogroup Icterohaemorrhagiae while an early MAT suggested Pomona. For this patient, a second MAT performed on a convalescent serum sample in routine eventually confirmed the serogroup Icterohaemorrhagiae. Table 4 shows the *lfb1* sequencing results and the details of serogroups.

## Comparison with the study conducted in the 2007–2014 period

Compared to the 2007–2014 period, the mean (SD) annual number of included cases reported to the population aged above 15 years tripled from 5.8 (2.7) to 14 (9.2) cases per year during the 2016–2022 period (p = 0.03, see Table 5). The 2016–2022 study population had more frequent comorbidities (35.6% vs. 15%, p < 0.01), despite a similar age. The rate of

**Table 4. Identified *Leptospira* species and presumptive serogroups of patients with leptospirosis in French Guiana, 2016-2022.**

| Number of cases | Species | *lfb1* | Serogroup |
|---|---|---|---|
| 15 | *Leptospira interrogans* | SG1 | Icterohaemorrhagiae |
| 1 | *Leptospira noguchii* | Undetermined | Undetermined |
| **Number of cases*** | | **Presumptive serogroup identified by MAT*** | |
| 22 | | Icterohaemorrhagiae | |
| 7 | | Canicola | |
| 1 per each | | Ballum, Panama, Pyrogenes | |
| 9 | | Coagglutinins | |

* Six patients with MAT analysis also had samples analyzed with *lfb1* sequencing.

** MAT was performed in routine with 11 serovar antigens panel in BIOMNIS lab and/or with the ancillary analysis with MAT with 23 serovar antigens panel at NRCL.

MAT: Microscopic Agglutination Test; NRCL: National Reference Center for Leptospirosis

**Table 5. Comparison of the main characteristics of patients with leptospirosis between the 2007-2014 and 2016-2022 periods, using the same case definition.**

| | 2007-2014 (N = 72) [3] | 2016-2022 (N = 188) | *p*-value* |
|---|---|---|---|
| Annual number of included cases, mean (SD) | 9.0 (4.6) | 26.9 (18.1) | – |
| Annual number of included cases per 100,000 of population aged > 15 years, mean (SD) | 5.8 (2.7) | 14.0 (9.2) | 0.03 |
| Age, median (IQR) | 39 (20-50.3) | 38 (28-52) | – |
| Age > 60 years old, N(%) | 9 (12.5) | 29 (15.4) | 0.45 |
| Comorbidities, N(%) | 11/69 (15.0) | 67/188 (35.6) | <0.01 |
| M/F sex ratio | 6.1 | 3.1 | 0.08 |
| Born in Brazil, N (%) | 24/66 (36.4) | 44/168 (26.2) | 0.10 |
| Born in Haiti, N (%) | 7/66 (10.6) | 49/168 (29.2) | <0.01 |
| Gold mining, N (%) | 12/48 (25.0) | 12/154 (7.8) | <0.001 |
| Suspicion of leptospirosis at first medical contact, N (%) | 14 (19.4) | 101 (53.7) | <0.001 |
| Positive PCR**, N (%) | 15 (20.8) | 102 (54.2) | <0.001 |
| In-hospital death, N (%) | 3 (4.9) | 4 (2.1) | 0.2 |
| ICU admission, N (%) | 12 (16.7) | 28 (14.9) | 0.90 |

* Student t-test was used for the comparisons of annual number of included cases, Chi 2 test was used for proportion comparison

** Blood, urine or CSF PCR

CSF: cerebrospinal fluid; ICU: Intensive care unit; M/F sex ratio: male/female sex ratio; PCR: Polymerase chain reaction.

male patients decreased (M/F sex ratio at 3.1 vs. 6.1, p = 0.08). There were also significantly more patients born in Haiti in the 2016–2022 study (29.2% vs. 10.6%, p < 0.01). Leptospirosis was suspected earlier and routine diagnosis was mostly based on PCR in the 2016–2022 study. Finally, the ICU admission rate and the in-hospital lethality rate were similar between the two periods.

## Discussion

### An increasing burden

Using the same inclusion criteria, the number of included cases was significantly greater for the 2016–2022 period than for the previous study that assessed the 2007–2014 period [3]. However, the rate of severe cases remained stable. This growing burden of the disease can be explained by several factors. First, as described previously [3,24],

the incidence of leptospirosis is linked to climatic events such as rainfall and flooding. Historical rainfalls in 2020, an especially 2021 and 2022 may have likely contributed to an increase in the number of cases, given that half the cases in our study were reported in 2021 and 2022. During these years the confinement probably resulted in the precarious population spending more time at home, in unsanitary and damp environment, as well as being more exposed to rodents at home. These factors could account for the increased number of cases in 2020, despite the lockdown. Another potential driving factor may be the increase in the number of people living in informal settlements in FG over the last decade [6], a well-known risk factor for leptospirosis [25]. In the present study, a third of the population with available data reported living in informal housing, which may be underestimate given the potential reluctance to disclose this information by the patient. Informal settlements may disrupt ecosystems due to human activity in forested areas, including within cities [26]. This can reduce biodiversity, which has been associated with an increased risk of leptospirosis transmission, as observed for other zoonotic diseases in French Guiana [27–28]. Third, the recent local studies on leptospirosis [3–4] may have contributed to a better awareness of the medical community towards leptospirosis and improved diagnostic capacity. This hypothesis is supported by a higher frequency of early clinical suspicion compared to the previous study.

### Change in patient profile

The global demographic profile of the patients was similar to that previously described in French Guiana and in the rest of the tropics [29–31] with a majority of young males. However, during the period 2016–2022, the proportion of males decreased, although not significantly, compared to the previous study in FG. It has been previously hypothesized that the principal reason for the predominance of male cases of leptospirosis may reflect the increased likelihood of males to have outdoor activities, especially at work [32]. However, a study in nearby Guyana during an outbreak of leptospirosis due to flooding showed a predominance of female cases [33]. Thus, considering that occupational risk factors were not predominant in our study, this could account for the rising proportion of female cases in recent years. Also, the lockdown resulting from the covid pandemic in 2020 might, associated to extreme climatic events, have led to a shift towards contamination at or near home. The study population was older than previously described, especially in the Guiana Shield and the Caribbean [25,34,35], with more people aged over 60 years and significantly more patients with comorbidities than in the 2007–2014 FG cohort. This may reflect an increase of elderly people living in conditions that increase the risk of leptospirosis. Better access to universal healthcare, which is possible in French Guiana even without health insurance [36], may have led to the identification of more comorbidities compared to the previous period.

The study population was mostly born in Haiti, which also differed from the 2007–2014 period where the most frequent place of birth was Brazil. This difference may reflect the lower proportion of gold miners, mostly originating from Brazil, in recent years among the leptospirosis cases. Furthermore, individuals born in Haiti constitute a significant group that experienced sustained migration in French Guiana during the 2015–2017 period and are exposed to higher levels of precariousness. In the present study, data on housing or health insurance coverage data were too limited to conduct further analysis on this factor especially among the population born in Haiti. This aspect should be especially a subject of particular interest in future studies.

When available, the data on health-insurance revealed a dramatically low coverage in the study population. This suggests that leptospirosis in French Guiana appears more as a disease of poverty and precariousness in French Guiana than a recreational disease as seen in other high-resource settings including mainland France [32–37]. It is noteworthy that during the last two years of the study, nearly half of the patients had no health insurance at all. This situation may have been exacerbated by administrative closures during the COVID-19 pandemic and by the increased vulnerability of the affected population.

## Exposure factors

In this study, the risk factor identified in patients was exposure to rodents, which is not surprising since it is the most widely recognized risk factor for *Leptospira* infection [30]. Proximity to open sewers or the accumulation of garbage, particularly in unstable housing conditions, amplifies this risk [38]. Environmental factors such as forest hiking and exposure to freshwater were also frequently reported in our population but without being able to precisely geolocate these exposures. The main professional exposure factors identified in the study (building work, gold mining, manual work) and fishing are known risk factors for leptospirosis in the region [39]. However, for many reasons, people with these occupations in FG may not benefit from preventive measures against leptospirosis (information and/or vaccination) which may be present in other better-recognized at-risk occupations such as slaughterhouse worker or garbage cleaner - barely reported in study patients. Interestingly, multiple exposures were reported in most of study patients even though some key factors -as flooding exposure -were rarely specifically asked. Furthermore, people living in precarious conditions (informal settlements) could also be exposed through at-risk occupations such as informal - and almost always manual- work which is commonly referred to as a 'job'. Thus, an incomplete anamnesis may lead to miss some of the exposure factors and possibly estimate an exposure as responsible while another one could be. This suggests that physicians should ask for many different sources of exposure to better estimate the risk of transmission near home, in bathing areas or hiking areas, or at the workplace given cumulative exposure is frequent. Indeed, many patients accumulated exposures from both inside and outside their home, raising the issue of the place of exposure, which is sometimes difficult to ascertain. This question is particularly relevant as human leptospirosis has been a notifiable disease in France since August 2023 [40]. Clinical signs and exposure factors of cases reported by doctors are now reported to the health authorities. The analysis of these data will allow us to identify epidemics and risk situations, thus increasing our epidemiologic knowledge of this infection in particular in French Guiana. Nevertheless, given the intercultural context of French Guiana and the profile of the population affected, data collection on exposure to leptospirosis and the geographical areas concerned would be best carried out by trained mediator-investigators [41]. This could improve the quality of surveillance data and optimize response measures.

## Identification of *Leptospira* strain and serogroup

As previously described [29], serogroup Icterohaemorrhagiae was predominant in our population, which is preferentially found in *Rattus* spp. This could reflect the precarious living conditions of the affected population and their proximity to waste and rodents. However, these data should be interpreted very cautiously as only few serogroups were identified in our study as MAT has not been routinely performed since 2014. Also, the difference in panels and laboratories performing the MAT in routine versus in the ancillary analysis is a limit to our study, even if there was only one discordance found, which was explained by the timing of the sample. In addition, *lfb1* sequencing results were obtained for a limited period of the study. Overall, this may have reduced the diversity of serogroups compared with the 2007–2014 period, when MAT data was available throughout the study period, and realized solely by the NRCL. For epidemiologic purposes, the infectious serogroup should be determined ideally with molecular tools which could be incorporated in routine diagnostics to help identify the infective strain and its associated serogroup [19,42].

## Weaknesses and strengths of the study

This study has several limitations. First, we used strict diagnostic criteria for leptospirosis diagnosed in FG hospitals, similar to the 2007–2014 study for comparison purposes. Indeed, we chose to maintain specific criteria to better assess confirmed or highly probable cases of leptospirosis, given the limitations of isolated positive IgM -especially moderately high serological titers - or intermediate levels of MAT results in an endemic area. This obviously led to an underestimate of the number of cases meeting these strict criteria. Thus, this study did not allow to estimate an accurate incidence. Furthermore, the case distribution and burden estimates presented here apply only to cases that met the

inclusion criteria and were treated in the hospital. Patients with true leptospirosis may have had missing data, isolated anti-Leptospira IgM serology with a coinfection or no criteria at all, due to the inappropriate timing of diagnostic tests and were not analyzed here [43]. During the first study 2007–2014 remote centers were included because the hospital center of Cayenne was already the referral center for remote health centers at this time and performed their biological analysis. Therefore, some patients were managed in remote health centers and transferred to Cayenne Hospital or with available medical information were included in the former study. Regarding the center of Kourou, no data was available for this center at the time of data collection for the 2007–2014 period. However, considering the limited number of cases in the 2016–2022 period for Kourou (n = 8) it is highly improbable that this could explain the increase of leptospirosis cases observed in the recent study. The rate of JHR was also probably underestimated, as only cases occurring outside the ICU and meeting the case definition were considered [44]. However, estimating the rate of JHR is always challenging in leptospirosis study [45]. Besides, despite systematic examination of all medical files, data were still missing on exposures and social coverage in many cases, due to either lack of investigation by the clinician or lack of report. Finally, due to the lack of realization of MAT testing in most cases and routine sequencing of the samples with positive PCR, data on serogroup were also scarce.

Despite its weaknesses, this study provides an important series with robust biological criteria similar to those of the previous study carried out in the same setting. The generalization of these findings is limited to French Guiana, but provides useful insights into recent trends in the local epidemiology.

## Conclusion

The global burden of leptospirosis is increasing in French Guiana and the characteristics of the affected population are evolving. In the recent period, multiple exposures were frequent with domestic and non-domestic risks. As in other tropical areas, leptospirosis particularly affects vulnerable urban populations. Public health measures should integrate these findings to better target at-risk populations and provide locally adapted leptospirosis information and prevention campaigns.

## Supporting information

**S1 Checklist. STROBE checklist.**
(DOCX)

**S1 Table. Collected variables.**
(DOCX)

**S2 Table. Biological characteristics of patients with leptospirosis in FG, 2016–2022 *.**
(DOCX)

**S3 Table. Details of biological diagnosis of the 188 study patients with leptospirosis.**
(DOCX)

## Acknowledgments

The authors would like to thank all the paramedics and doctors from different specialties and from all the hospitals in French Guiana who managed the patients in this study, as well as all the biologists in charge of microbiological or molecular diagnosis. The authors especially acknowledge Fabrice Quet (CIC Inserm 1424, Cayenne University Hospital, Cayenne 97300, French Guiana) for his help in providing the number of FG population data and Pierre-Yves Carlier (freelance journalist, Cayenne, French Guiana) for his help in providing the number of hospital beds. The authors also thank the staff of the French National Reference Center for Leptospirosis for performing MAT and *lfb1* typing for some of the samples.

## Author contributions

**Conceptualization:** Mathilde Zenou, Céline Michaud, Paul Le Turnier.

**Data curation:** Mathilde Zenou, Pascale Bourhy, Arsène Kpangon, Jean-François Carod, Christelle Prince, Sabine Trombert-Paolantoni, Alexia Barbry, Paul Le Turnier.

**Formal analysis:** Mathilde Zenou, Mathieu Nacher, Paul Le Turnier.

**Methodology:** Mathilde Zenou, Loïc Epelboin, Paul Le Turnier.

**Supervision:** Pascale Bourhy, Loïc Epelboin, Paul Le Turnier.

**Writing – original draft:** Mathilde Zenou, Loïc Epelboin, Paul Le Turnier.

**Writing – review & editing:** Mathilde Zenou, Pascale Bourhy, Philippe Abboud, Mona Saout, Félix Djossou, Céline Michaud, Arsène Kpangon, Alexis Fremery, Nicolas Higel, Jean-François Carod, Christelle Prince, Sabine Trombert-Paolantoni, Alexia Barbry, Mathieu Nacher, Mathieu Picardeau, Loïc Epelboin, Paul Le Turnier.

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
