## [Decision Letter · Decision Letter 0]

13 Aug 2025

PNTD-D-25-00800

Evolution of human leptospirosis in French Guiana, 2016-2022

Dear Dr. Le Turnier,

Thank you for submitting your manuscript to PLOS Neglected Tropical Diseases. After careful consideration, we feel that it has merit but does not fully meet PLOS Neglected Tropical Diseases's publication criteria as it currently stands. Therefore, we invite you to submit a revised version of the manuscript that addresses the points raised during the review process.

Please submit your revised manuscript within 60 days Oct 12 2025 11:59PM. If you will need more time than this to complete your revisions, please reply to this message or contact the journal office at plosntds@plos.org. Please include the following items when submitting your revised manuscript:

We look forward to receiving your revised manuscript.

Kind regards,

Stuart Blacksell

Section Editor

Shaden Kamhawi

co-Editor-in-Chief

Paul Brindley

co-Editor-in-Chief

**Additional Editor Comments :**

1. General Comments - This manuscript presents potentially valuable data on leptospirosis in French Guiana; however, in its current form, it requires substantial revision to ensure clarity, consistency, and scientific robustness. The reliability of the conclusions is wholly dependent on the quality of the diagnostics used. The use of multiple diagnostic approaches raises important questions about comparability, consistency, and the overall confidence in the results.

2. Clarity and Structure of Diagnostic Descriptions - The description of the diagnostic methods is fragmented and presented out of sequence, which makes it difficult for the reader to follow. For example, the microscopic agglutination test (MAT) is first mentioned on line 148 but only fully introduced on line 165. Similarly, diagnostic cut-off values for MAT are discussed before the leptospira serovars used in the assay are described. For clarity, each diagnostic method should be presented in a consistent order: define the test and its abbreviation at first mention, describe the methodology, outline cut-off values, and then explain the rationale for these parameters. Abbreviations should be introduced once and then used consistently throughout.

3. MAT - The rationale for the selection of leptospira serovars used in the MAT is not explained. Line 166 states that 11 antigens were used routinely in “Biomnis lab,” but it is unclear what this laboratory is and why these serovars were chosen. Was this decision informed by previous local isolation of invasive Leptospira strains, historical epidemiological data, or another factor? In addition, it is not clear whether serovars were tested individually or in pooled form. The description in lines 154–156 suggests individual testing in some cases, but this should be stated explicitly.

Also, the diagnostic thresholds used for case classification require justification. The definitions of “certain” (≥1:400) and “probable” (1:200) cases given in lines 147–149 should be referenced to relevant guidelines or prior literature. Similarly, the classification of probable cases based solely on anti-Leptospira IgM levels above 50 IU/ml (lines 149–152) without positive MAT or PCR requires explanation. Please confirm whether this was an ELISA and provide further detail on the assay itself.

4. PCR Testing – The presentation of the PCR data contains several inconsistencies that need resolution. In the results (line 342), it is stated that PCR diagnosis was performed in 143 patients, but the flow chart in Figure 1 shows 1797 patients tested, with 117 confirmed cases by PCR. These figures appear contradictory and should be reconciled. Additionally, the Materials and Methods section should clearly describe the types of samples collected for PCR, the justification for using different sample types, and the timing of collection in relation to illness onset. This is important, as timing can have a significant effect on positivity rates. In the results (lines 342–345), findings from different sample types are presented, but this information should be introduced earlier in the methods to provide context.

5. Serology – Lines 346–347 suggest that acute and convalescent sampling was performed, with first MAT tests conducted in 51/188 patients and a second in only 7/51. If these represent acute and convalescent pairs, please provide details on the number of days from illness onset to the first sample, the interval between acute and convalescent collections, and any reasons for the low rate of paired sampling. As with the PCR data, ensure that these figures align with the flow chart in Figure 1.

6. Additional Considerations - It would strengthen the study to state whether in vitro isolation of Leptospira was attempted to confirm infecting serovars. This would provide an additional level of diagnostic confirmation and improve confidence in the epidemiological conclusions. Finally, the manuscript would benefit from careful language editing to improve clarity and readability, as well as to ensure that terminology is used consistently.

**Journal Requirements:**

1) Please upload all main figures as separate Figure files in .tif or .eps format. For more information about how to convert and format your figure files please see our guidelines: 

2) We notice that your supplementary Table (Table S2 ) is included in the manuscript file. Please remove it and upload it with the file type 'Supporting Information'. Please ensure that each Supporting Information file has a legend listed in the manuscript after the references list.

3) The following file is currently uploaded as file type 'Other', which is not viewable by the reviewers: STROBE checklist EVOLEPTO.docx. Please change the file type to 'Supporting Information' and include a legend in the manuscript if you wish it to be included in review.

**Comments to the Authors: **

**Please note that one of the reviews is uploaded as an attachment.**

**Reviewers' Comments:**

Reviewer's Responses to Questions

**Key Review Criteria Required for Acceptance?**

**Methods**

-Are the objectives of the study clearly articulated with a clear testable hypothesis stated?

-Is the study design appropriate to address the stated objectives?

-Is the population clearly described and appropriate for the hypothesis being tested?

-Is the sample size sufficient to ensure adequate power to address the hypothesis being tested?

-Were correct statistical analysis used to support conclusions?

-Are there concerns about ethical or regulatory requirements being met?

Reviewer #1: The study presents clear objectives aimed at updating the epidemiological and clinical characteristics of leptospirosis in French Guiana. While the hypothesis is not explicitly stated, the research questions and comparisons with previous findings suggest a well-defined investigative framework. The study design, incorporating multiple hospitals and a remote clinic, is appropriate for addressing these objectives, though variations in case identification methods between the two periods pose challenges for direct comparisons. Adjustments for population size may be necessary to ensure adequate statistical power for comparisons. Most statistical analyses appear robust; however, concerns arise regarding the use of raw case counts without adjusting for population differences. There are no concerns regarding ethical procedures.

Reviewer #2: okay

Reviewer #3: yes

**Results**

-Does the analysis presented match the analysis plan?

-Are the results clearly and completely presented?

-Are the figures (Tables, Images) of sufficient quality for clarity?

Reviewer #1: The analysis and results align with study aim. All tables need improvement.

Reviewer #2: okay

Reviewer #3: Yes

**Conclusions**

-Are the conclusions supported by the data presented?

-Are the limitations of analysis clearly described?

-Do the authors discuss how these data can be helpful to advance our understanding of the topic under study?

-Is public health relevance addressed?

Reviewer #1: The conclusion is shalow and the implications to public health is not properly addressed.

Reviewer #2: Results of the study are limited to the small study area in French Guiana

Otherwise okay

Reviewer #3: yes

**Editorial and Data Presentation Modifications?**

Reviewer #1: (No Response)

Reviewer #2: (No Response)

Reviewer #3: No modifications required

**Summary and General Comments**

Reviewer #1: This study provides a comprehensive update on the epidemiological and clinical characteristics of leptospirosis in French Guiana. It is highly relevant, however, the presentation of results is somewhat disorganised, and the tables require significant improvements.

I have a major concern regarding the statistical robustness of the comparison between the 2007-2014 study and the current study. Alone, the results from the 2016-2022 study are extensive and valuable. The effort of comparing two comprehensive research studies in the same country is valid. However, direct comparisons between the two periods must be approached with caution due to differences in data sources. Although the authors claim to have used similar case definitions in both studies, the manner in which cases were obtained differs. The earlier study (2007–2014) included only two hospitals, while the current study expands to three hospitals and a remote clinic centre. Given these discrepancies, even the total number of cases should be adjusted based on population size to improve the statistical power of comparisons.

When comparing the number of cases per year between the two studies, a Student's t-test could be appropriate if the data meet the assumptions of normality and equal variance. However, raw case counts alone do not account for population differences, meaning a direct comparison may be misleading.

Reviewer #2: The authors present results of serological surveys in French Guiana. The work is well done, but is of little significance to anyone outside of the study area.

Minor concern: the use of "evolution" in the title is not appropriate, since there is no evidence that the disease evolved.

Reviewer #3: This study presents an updated analysis of leptospirosis surveillance data in French Guiana over a seven-year period (2016–2022), with comparisons to a prior dataset from 2007–2014. The authors report shifts in the geographic distribution of cases, a modest increase in the median age of affected individuals, continued male predominance, and greater use of PCR diagnostics. Seasonal patterns remain, but occasional off-season clusters were detected. The manuscript addresses a neglected tropical disease in a high-incidence region, and the comparison with historical data adds value.

1. The study compared data from 2016–2022 with 2007–2014. Were the case definitions, diagnostic methods, and inclusion criteria identical across periods?

2. The study reports seasonal peaks during the rainy season. Has it been tested statistically, and did you consider other environmental variables such as flooding or river levels?

3. The study did not present any serovar distribution. Without the serovar information, is it possible to identify reservoirs and tailor interventions?

4. Lines 139 to 143

Inclusion criteria > 15 years at diagnosis, and exclusion criteria <18 years at the time of the data collection.

What was the minimum age of the patient to be included in the study?

5. Lines 157-159

PCR diagnosis was based on an in-house real-time PCR targeting the 16S RNA coding

gene and lipL32 gene. All PCR were performed by Biomnis laboratory (reference RIHN :

N131).

Any reference for the PCR. Are the amplicons sequence confirmed.

6. Line 258

One hundred fifty-five patients (82.5%) patients initially

consulted in CHC

Why majority of the cases in CHC ?

7. Figure 2: Why there is a high incidence in 2021? Was this linked to any specific outbreak or environmental factors?

8. Year 2020, during the era of covid, the cases were still same as year 2019 or 2022, despite of lockdowns and quarantines. So even occupational exposure, the cases did not go down, is it something to do with the living sanitation to present rodents exposure.

9. Line 359 * Six patients with MAT analysis also had samples analyzed with lfb1 sequencing. Was the results same for the serogroup

PLOS authors have the option to publish the peer review history of their article (what does this mean? ). If published, this will include your full peer review and any attached files.

**Do you want your identity to be public for this peer review?** For information about this choice, including consent withdrawal, please see our Privacy Policy .

Reviewer #1: No

Reviewer #2: No

Reviewer #3: No

**Figure resubmission:**
---

## [Editor Report · Decision Letter 1]

5 Oct 2025

Dear Dr. Le Turnier,

We are pleased to inform you that your manuscript 'Evolution of human leptospirosis in French Guiana, 2016-2022' has been provisionally accepted for publication in PLOS Neglected Tropical Diseases.

Best regards,

Joseph M. Vinetz

Section Editor

Joseph Vinetz

Section Editor

Shaden Kamhawi

co-Editor-in-Chief

Paul Brindley

co-Editor-in-Chief

---

## [Editor Report · Acceptance letter]

Dear Dr. Le Turnier,

We are delighted to inform you that your manuscript, "Evolution of human leptospirosis in French Guiana, 2016-2022," has been formally accepted for publication in PLOS Neglected Tropical Diseases.

Best regards,

Shaden Kamhawi

co-Editor-in-Chief

Paul Brindley

co-Editor-in-Chief
